# Separation of Fructosyl Oligosaccharides in Maple Syrup by Using Charged Aerosol Detection

**DOI:** 10.3390/foods10123160

**Published:** 2021-12-20

**Authors:** Kanta Sato, Tetsushi Yamamoto, Kuniko Mitamura, Atsushi Taga

**Affiliations:** Pathological and Biomolecule Analyses Laboratory, Faculty of Pharmacy, Kindai University, 3-4-1 Kowakae, Higashiosaka 577-8502, Japan; kantasato713@gmail.com (K.S.); yamatetsu@phar.kindai.ac.jp (T.Y.); mitamura@phar.kindai.ac.jp (K.M.)

**Keywords:** HPLC, charged aerosol detection, CAD, fructosyl oligosaccharide, FOS, maple syrup, neokestose, 1-kestose, nystose

## Abstract

Fructosyl oligosaccharides, including fructo-oligosaccharide (FOS), are gaining popularity as functional oligosaccharides and have been found in various natural products. Our previous study suggested that maple syrup contains an unidentified fructosyl oligosaccharide. Because these saccharides cannot be detected with high sensitivity using derivatization methods, they must be detected directly. As a result, an analytical method based on charged aerosol detection (CAD) that can detect saccharides directly was optimized in order to avoid relying on these structures and physical properties to clarify the profile of fructosyl oligosaccharides in maple syrup. This analytical method is simple and can analyze up to hepta-saccharides in 30 min. This analytical method was also reliable and reproducible with high validation values. It was used to determine the content of saccharides in maple syrup, which revealed that it contained not only fructose, glucose, and sucrose but also FOS such as 1-kestose and nystose. Furthermore, we discovered a fructosyl oligosaccharide called neokestose in maple syrup, which has only been found in a few natural foods. These findings help to shed light on the saccharides profile of maple syrup.

## 1. Introduction

Maple syrup is a popular natural sweetener made by boiling sap from sugar maple trees. As the season progresses, maple syrup’s color darkens, and it is classified into four grades: golden, amber, dark, and very dark [1]. Maple syrup has various biological effects, and it has also been reported that different grades of maple syrup could have different biological effects [2,3,4,5,6]. Although maple syrup has a low glycemic index (GI), it is mostly sucrose with a small amount of glucose and fructose [5,7,8,9]. It was reported that maple syrup contains pentoses, such as xylose, arabinose, and ribose, and polysaccharides, such as inulin [10,11]. Our previous study discovered that maple syrup contains a fructosyl oligosaccharide known as blastose [12]. Furthermore, this study also suggested the presence of an unidentified compound in maple syrup that was considered to be fructosyl oligosaccharide with a polymerization degree of at least three.

Recently, fructosyl oligosaccharides have gained popularity, with a 1-kestose, nystose and 1-fructofuranosylnystose serving as examples (Figure 1). These saccharides are known as fructo-oligosaccharides (FOS); they have been reported to have functions such as indigestibility and prebiotics, which are beneficial to human health [13,14]. Although these saccharides have been found in natural products such onion, burdock, and banana [15,16,17,18,19], the amounts are minute. It is critical to quantify FOS in order to make effective use of natural products containing FOS as functional foods. However, due to the capping of the aldose-reducing terminal by fructose, FOS cannot be analyzed with high sensitivity using derivatization methods. As a result, there is a need for an analytical approach with high sensitivity that can directly analyze FOS.

Charged aerosol detection (CAD), refractive index detection (RID), and pulsed amperometric detection (PAD) are analytical methods for saccharide detection [20,21,22]. Although RID is used for routine analysis and the analysis of known compounds, it has been reported that the LOQ for saccharides analysis by HPLC-RID is in the 0.18–0.30 g/L range [22]. On the other hand, PAD can analyze several saccharides with high resolution; it requires an anion exchange column and sodium hydroxide as the mobile phase. Because desalting is required, it is challenging to identify unidentified compounds in natural products. As a solution to these issues, CAD has become widely used in recent years. CAD is more versatile and has a more comprehensive detection range than RID. Its capabilities include detecting a wide range of semi- and non-volatile compounds such as saccharides but also lipids, amino acids, and peptides [23,24,25,26]. Its responsiveness is independent of the analyte’s molecular structure or physical properties. Therefore, unknown compounds can be quantified using known compounds [27]. These properties are ideal for quantifying neutral saccharides with no UV absorption and analyzing compounds that cannot be derivatized, such as FOS. Because gradient elution for CAD is possible, it is possible to analyze unidentified oligosaccharides that are likely to be overlooked by RID methods in the analysis of natural products. As a result, CAD can detect trace amounts of unidentified oligosaccharides found in natural products. Although several CAD applications for analyzing aldoses such as glucose have been reported [20,23], it has not been reported for analyzing saccharides fructosyl oligosaccharides including FOS.

This study aims to optimize the analysis method using FOS standards by hydrophilic interaction chromatography (HILIC)-CAD to clarify the profile of fructosyl oligosaccharides, including FOS in maple syrup. This optimized method is simple, compact, and reliable, and reproducible. While the RID method could only detect sucrose, glucose, and fructose, this optimized method using CAD was able to observe the several rare saccharides as well as the major saccharides such as sucrose. In addition, its quick application was used to identify and quantify several FOS in maple syrup precisely. Furthermore, we discovered that maple syrup contained one FOS, which is rarely reported in natural foods.

## 2. Materials and Methods

### 2.1. Reagents and Chemicals

Fructose, glucose, and sucrose were purchased from Nacalai Tesque, Inc. (Kyoto, Japan). Nystose, 1-kestose, and 1-fructofuranosylnystose were purchased from FUJIFILM-Wako Pure Chemicals Co (Osaka, Japan). Maple Farms Japan, Inc. (Osaka, Japan) generously provided all maple syrup of grade A and sap. These maple syrups are made from sugar maple (*Acer saccharum*) and produced by Bascom Maple Farms, Inc. (Acworth, NH, USA). All other chemicals were of the highest purity and obtained from Nacalai Tesque, Inc. (Kyoto, Japan).

### 2.2. Sample Preparation

All saccharide standards were prepared at the following concentrations: 0.5, 1.0, 5.0, 10, 25, 50, 75, and 100 μg/mL. The maple syrup was dried and then dissolved in water. This solution was centrifuged for 5 min at 3000× *g*. The supernatant was filtered via a 0.45 μm filter (Merck Millipore, Darmstadt, Germany), and acetonitrile was added to the filtrate to adjust the solvent concentration to 75% acetonitrile.

### 2.3. Analysis by High-Performance Liquid Chromatography with CAD (HPLC-CAD)

Maple syrup samples were analyzed using an LC 10 Advp system (Shimadzu Corporation, Kyoto, Japan) equipped with a Corona Veo detector (Thermo Fisher Scientific, Inc., Waltham, MA, USA). Separation was performed using an Asahipak NH2P-50 4E column (5 µm, 4.6 mm internal diameter ×250 mm, Showa Denko K.K., Tokyo, Japan) as an HILIC column. The mobile phase consisted of deionized water (solvent A) and acetonitrile (solvent B) with multi-gradient elution in 75% solvent B from 0 to 10 min and 75 to 50% B from 10 to 30 min. At room temperature (~23 °C), the flow rate of 1 mL/min. The injection volume was set to 20 µL. The data processing was carried out using Chromeleon 7.2.2 software (Thermo Fisher Scientific, Massachusetts, MA, USA). The pure nitrogen gas for the CAD was flowed by an AT-2NP-CAD (AIR-TEC Co., Kanagawa, Japan). The sampling rate was set to 5.0 Hz with a filter constant of 5.0 s. The response and signal correction power function was set to 1.0. The fractionate of a target oligosaccharide was carried out using an Asahipak NH2P-50 10E column (5 µm, 10.0 mm internal diameter ×250 mm; Showa Denko K.K.) with 75% acetonitrile as the mobile phase was. A flow rate of 2 mL/min was used for isocratic elution with Adjustable Flow Splitter (Thermo Fisher Scientific Inc., Waltham, MA, USA) as a post-column type splitter. The split ratio was 1:20, with the low-flow outlet directed to CAD. The remaining (~95%) volume was then collected from the splitter’s high-flow outlet. The other conditions were as previously described.

### 2.4. HPLC-CAD Method Validation

Based on the International Council for Harmonization and Technical Requirements (ICH) guidelines [28], the developed HPLC-CAD method was validated in terms of concentration range, linearity, limit of detection (LOD), limit of quantitation (LOQ), precision, and recovery. All experiments were performed at least three times.

#### 2.4.1. Linearity

The linearity experiments were performed with eight different concentrations (0.5–100 µg/mL) of standards. Each calibration curve was fitted to a linear, quadratic, or logarithmic equation, and each equation’s correlation coefficient was calculated. Additionally, the concentration error was calculated by fitting the area value of eight standards to each equation.

#### 2.4.2. LOD and LOQ

The LOD and quantification (LOQ) were estimated in accordance with ICH guidelines [28]. LOD and LOQ are defined as a signal-to-noise ratio of 3:1 and 10:1, respectively. 

#### 2.4.3. Precision

The repeatability of the analytical method was calculated to assess its precision. The precision was measured in terms of the relative standard deviation (RSD, %). Equation (1) was used to calculate the RSD of the area value obtained from each standard of 50 μg/mL. Furthermore, intra-day and inter-day precision were obtained by consecutively injecting nine working standards on the same day and three working standards on different days.
(1)$$\mathrm{RSD}\;\left(\%\right)=\left(\frac{\mathrm{Standard}\;\mathrm{Deviation}}{\mathrm{Mean}}\right)\;\times \;100$$

#### 2.4.4. Accuracy

To investigate the recovery and validity of the analytical method, the recovery was calculated using Equation (2). FOSs, monosaccharides, di-saccharides standards (50 µg/mL) were added 10 mg, 1 mg, and 100 μg/mL very dark-grade maple syrup samples, respectively. The recovery was then calculated from the area values of the added sample, standard product, and maple syrup sample using the following equation.
(2)$$\mathrm{Recovery}\;\left(\%\right)=\left(\frac{\mathrm{Observed}\;\mathrm{conc}.\;-\;C_{\mathrm{maple}}}{\mathrm{Added}\;\mathrm{conc}.}\right)\;\times \;100$$
where C_maple_ is the determined concentration of saccharides in maple syrup.

### 2.5. Size Exclusion Chromatography

The excluding of the large-molecular-weight components in the maple sap was carried out by Ultracel 10 kDa ultrafiltration discs (Merck Millipore, Darmstadt, Germany). To separate the target oligosaccharide and other saccharides in maple sap, the filtrate was applied on an Econo-column (1 m × 25 mm internal diameter; Bio-Rad Laboratories, Inc., Hercules, CA, USA) packed with Sephadex G-15 gel (GE Healthcare, Boston, MA, USA). A Bio-Rad fraction collector Model 2110 (Bio-Rad Laboratories, Inc.) was used to collect each fraction, which consisted of 200 drops. Water was used for elution and washing.

### 2.6. Nuclear Magnetic Resonance (NMR)

The structural analysis of the target oligosaccharide was carried out using a JNM-ECA 800 model (Jeol Resonance, Inc., Tokyo, Japan) and ^1^H and ^13^C-NMR spectroscopic data were recorded at 800 and 200 MHz, respectively. The unidentified oligosaccharide for NMR analysis was dissolved in D_2_O. Radiant-selected pulse sequences were used to ^1^H–^1^H correlation spectroscopy (COSY), ^1^H–^13^C heteronuclear single-quantum correlation spectroscopy (HSQC), ^1^H–^13^C heteronuclear multiple-bond correlation spectroscopy (HMBC) and ^1^H–^1^H nuclear overhauser effect spectroscopy (NOESY) spectra. Chemical shifts are given in the δ-scale (ppm), and coupling constants *J*
_H,H_ are given in Hz.

### 2.7. Statistical Analysis

The statistical analyses were performed using Microsoft Office Excel 365 (Redmond, Washington, WA, USA). All experiments were performed at least three times. All data are expressed as the mean ± standard deviations (S.D.) of the mean.

## 3. Results

### 3.1. Developments of the Method for the Separation of FOSs

To analyze FOS using HPLC with CAD, the separation method was optimized using glucose, fructose, sucrose, and FOSs (1-kestose, nystose, and 1-fructofuranosylnystose) standards (Figure 1). At room temperature, isocratic runs were investigated at a flow rate of 1 mL/min flow with acetonitrile:water ratios of 60:40, 65:35, 70:30, 75:25, and 80:20 were investigated (Appendix A). When performing the isocratic elution with acetonitrile at a concentration of 70% or higher, good resolution between each standard was obtained. 

Subsequently, to compactly separate FOSs while maintaining the separation of mono- and disaccharides, we decided to perform gradient elution after 10 min. The mobile phase consisted of a 70:30 isocratic ratio for 10 min, followed by a 70% to 50% acetonitrile linear gradient for 20 min (Figure 2A). A linear gradient elution had a slight effect on the 1-kestose and nystose peaks, but it did shorten the retention time of the 1-fructofuranosylnystose peak. The retention times for 1-kestose, nystose, and 1-fructofuranosylnystose were significantly reduced for a linear gradient of 75% to 50% acetonitrile (Figure 2B). In addition, the shapes of these peaks were improved. The retention times of the sucrose, 1-kestose, nystose, and 1-fructofranosylnystose peaks were significantly reduced in linear gradients of acetonitrile from 80% to 50%. The peak retention times for 1-kestose, nystose, and 1-fructofuranosylnystose were 24.2 min, 26.4 min, and 28.1 min, respectively. Tri-, tetra-, and penta-saccharides were retained for about 4 min, narrowing the separation window (Figure 2C). These findings indicated that the multi-gradient method using 75% acetonitrile as the starting eluent can be used to analyze mono- to penta-saccharides including FOSs uniformly with high resolution. Therefore, since it may provide sufficient resolution for the analysis of small oligosaccharides as well as fructosyl oligosaccharides in maple syrup, isocratic gradient elution with 75% acetonitrile was selected as the optimal condition. On the other hand, the polymerization degree of oligosaccharides that could be analyzed by this analytical method was investigated. This analytical method was able to analyze up to hepta-saccharides 30 min after analyzing inulin oligosaccharides (Appendix A).

### 3.2. Validation of the Optimized Method

The nonlinear calibration range, correlation coefficient (R^2^), LOD, LOQ, intra-day precision, inter-day precision, and recovery of this analytical method were all evaluated. Table 1 summarizes the data pertaining to method validation. Each quantification range for all standards includes 0.5–100 μg/mL, with three digits including LOQ. It can be presented as a quadratic equation y = ax^2^ + bx + c from the responses of evaluated ranges. The correlation coefficient (R^2^) expressed linearity of at least R^2^ > 0.999 for all saccharides. The LOD and LOQ for all standards were 0.25 μg/mL and 0.5 μg/mL, respectively. Precision was 0.9% or less intra-day and 1.8% or less inter-day. The recovery of all standards was calculated using the maple syrup sample described below, and good values of 91% or higher were obtained for all standards. 

### 3.3. Analysis of Carbohydrates Observed in Maple Syrup

This optimized method was compared to the RID method by analyzing maple syrup samples. Figure 3 showed the carbohydrates found in very dark-grade maple syrup using RID and CAD. The mobile phase consisted of a 75:25 isocratic ratio for 10 min, followed by a 75% to 50% acetonitrile linear gradient for 20 min. As a result, only major saccharides such as sucrose were observed by the RID method. However, due to the baseline drift caused by linear gradient elution, peaks could not be observed after sucrose (Figure 3A). On the other hand, in the CAD method, for monosaccharides, the peak of psicose was observed, which could not be derivatized, as well as fructose. This optimized method separated the saccharides found in maple syrup into a tetra-saccharide. The peaks of 1-kestose and nystose were observed, but 1-fructofuranosylnystose was not found in maple syrup (Figure 3B).

On the other hand, any unidentified peaks were observed at 18–21 min of this chromatogram, which was predicted to be a trisaccharide based on the retention time. These unidentified saccharides were named “mapletrioses”. Mapletrioses were identified as three distinct peaks and were numbered in ascending order of retention time. Mapletriose1 was the fourth highest peak in maple syrup after sucrose, glucose, and fructose, and it slightly overlapped with the peak of 1-kestose. Mapletriose1 and 1-kestose had a resolution of 1.4. On the other hand, mapletriose2 and 3 were present in trace amounts. Although we attempted to identify mapletrioses using standards available on the general reagent market, these saccharides were unable to be identified.

Subsequently, saccharides quantification in all grades of maple syrup was performed using standard calibration curves (Table 2). The amount of fructose and glucose in maple syrup increased as the season progressed. Though sucrose amount did not change significantly by well controlled brix, the sucrose amount decreased to approximately 55% in the very dark grade. Additionally, 1-kestose and nystose levels increased as the season progressed, as did monosaccharides. On the other hand, we also attempted to quantify mapletriose1 from its area value using the 1-kestose calibration curve. The predictive quantity of mapletriose1 increased with the late season, and each grade maple syrup contained 20.6, 19.6, 31.2, and 40.6 μg/10 mg. Since the predicted values of mapletriose1 in the saccharides composition of maple syrup were not negligible, we decided to identify it.

### 3.4. Structural Analysis of Mapletriose1 by NMR

For structural analysis of mapletriose1, this oligosaccharide was fractionated and purified from maple sap in accordance with our previous study [12]. Two-dimensional NMR analysis was performed to clarity the structure of mapletriose1. The C-2 and C-2″ of fructose residues were observed in the one-dimensional carbon spectrum at δ_C_ 106.42 ppm and δ_C_ 106.43 ppm, respectively. In the one-dimensional proton spectrum, the H-1′ of glucoside residue resonated at δ_H_ 5.39 ppm with a *J*
_H,H_ value of 3.9 Hz. In addition, the H-6′ of glucoside residue was identified at δ_H_ 3.77–3.79 ppm from the correlation spectrum COSY, HSQC, and DEPT. These proton signals (H-1 and H-6 of glucose residue) correlated with fructofuranoside C-2 and C-2″ position (δ_C_: 106.42–106.43) in the two-dimensional HMBC spectrum. Protons corresponding to the H-4 fructose residue showed correlations to the proton H-1′ of a glucose residue in the two-dimensional NOESY spectrum. All of the other signals from one-dimensional and correlation spectroscopy (i.e., COSY, HSQC, HMBC, NOESY, and DEPT) were assigned as a group (Table 3, Appendix A). Based on these findings, mapletriose1 is predicted to be a trisaccharide called neokestose (2,6-*O*-β-d-fructofuranosyl-α-d-glucopyranosyl-1,2-*O*-β-d-fructofuranoside), and its structural formula is shown in Figure 4.

## 4. Discussion

An HILIC column is commonly used for saccharide analysis. In this study, an amino-type HILIC column was used. Its distinguishing feature is the separation of saccharides based on the degree of polymerization. The retention time can, thus, be used to predict the degree of polymerization of unidentified saccharides. However, using only isocratic elution, it is difficult to rapidly analyze the saccharide with a degree of polymerization of three or more. As a result, multi-gradient elution was used to analyze these saccharides rapidly and with high sensitivity. If highly polymerizable saccharides or mono- and di-saccharides are expected to be present in samples, it may be preferable to analyze them with acetonitrile at 70 and 80% concentrations, respectively. However, because our target in this study was fructosyl oligosaccharides in maple syrup, we used 75% acetonitrile isocratic elution for 10 min followed by linear gradient to 50% acetonitrile. In addition, inulin was analyzed using this optimized method, and up to hepta-saccharides could be analyzed within 30 min, which is comparable to the analysis time of PAD-based methods specialized for saccharides analysis [9,29,30,31]. Furthermore, we validated this optimized method and demonstrated its reliability with good results. Therefore, this optimized method is also suitable for routine analysis of fructosyl oligosaccharides such as FOS.

Subsequently, we used this optimized method to analyze saccharides in maple syrup. In the current study, we found 1-kestose, nystose, and several unidentified tri-saccharides referred to as “mapletrioses” from maple syrup based on the carefully analyzed FOSs of maple syrup. Recent studies have analyzed the saccharides in maple syrup using analytical methods using PAD, but there are no reports of these saccharides [9]. Because 1-kestose is the smallest unit of inulin and nystose is fructosylated 1-kestose, these results support previous studies that suggested that maple syrup contained inulin [11]. Our previous studies could suppress the rise in plasma glucose levels [2,3]. This function became increasingly noticeable as the season progressed. Because maple syrup contains various saccharides, including fructosyl oligosaccharides, we hypothesized that this function is related to the saccharide content of maple syrup. As a result of quantifying FOSs in maple syrup in the current study, it was discovered that FOSs contents increased as the season progressed, as with maple syrup’s function. These FOS have been linked to the indigestible function [14]. Therefore, the content of these FOSs may be related to maple syrup’s function in suppressing the increase in plasma glucose level.

Our previous study found blastose in maple syrup [12]. This study also showed that blastose levels were rising due to invertase digestion of maple syrup. Invertase, also known as β-fructofuranosidase, can cleave a glycosyl bond between fructose and glucose from the fructose side. This occurrence suggested that maple syrup contained an unidentified fructosyl oligosaccharide that was considered to be fructosyl blastose. On the other hand, blastose is a disaccharide composed of fructose and glucose with a β(2→6) linkage, and its structure is not shared by 1-kestose or nystose. As a result of our prediction that a mapletriose is the fructosylated blastose, we set out to isolate and purify mapletriose1 from maple sap to determine its structure. The purified and isolated mapletriose1 was analyzed by LC-ESI-MS/MS, and m/z of this saccharide was 503.1 as [M-H]^−^. This was consistent with the predicted mass of tri-saccharides. Furthermore, NMR analysis was performed to analyze the detailed structure of mapletriose1, including the linkages of its constituent saccharides. NMR spectroscopy showed that mapletriose1 is a fructosyl blastose called neokestose composed of two fructose molecules and glucose. Neokestose is also known as neo-inulin type fructosyl oligosaccharide or neo-FOS. While it has been established that neokestose is a transfructosylation product of levansucrase [32,33], little is known about its quantitative composition in natural foods because its standard is not available on the general reagent market. 

In blastose, C-2 of fructose is linked to C-1 of glucose, a structural feature of neokestose (Appendix A). In addition, blastose has been reported to inhibit glycosidase activity. Because the structure of neokestose includes blastose, a functional oligosaccharide, we expect neokestose to have some function similar to blastose. As a result, in future research, we should examine the function of neokestose. Furthermore, we believe that clarifying this observation will lead to the detailed mechanism by which maple syrup suppresses the rise in plasma glucose level. 

In contrast, four unidentified peaks, including mapletriose2 and 3, were confirmed in the current study. It is expected that qualitative analysis will be complicated without standards. However, further studies are needed to identify these peaks via isolation, purification, and structural analysis of these compounds to reveal the saccharide profile of maple syrup fully.

## 5. Conclusions

In this study, we used CAD to optimize the analysis methods for fructosyl oligosaccharides in order to clarify the saccharide profile in maple syrup. This method is simple and compact, and it can analyze up to hepta-saccharides within 30 min. In addition, we used this analytical method to precisely determine the mono- and di-saccharides and fructosyl oligosaccharides, including FOS, in maple syrup of all grades. Maple syrup contained sucrose, glucose, and fructose as major saccharides; it also contained FOS such as 1-kestose and nystose. Furthermore, we discovered a fructosyl oligosaccharide called neokestose in maple syrup. This fructosyl oligosaccharide is rarely reported in natural foods. Because of their structural characteristics, these fructosyl oligosaccharides, including FOSs, may be related to the function of maple syrup.

## Figures and Tables

**Figure 1 foods-10-03160-f001:**
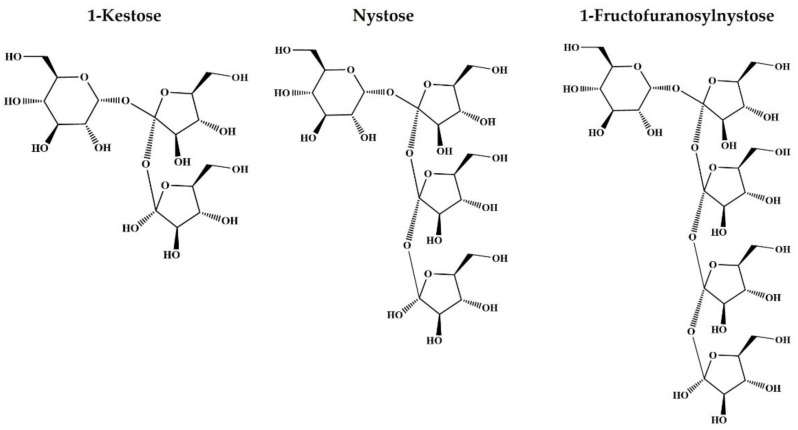
The structures of 1-kestose, nystose, and 1-fructofuranosylnystose as the representative FOS.

**Figure 2 foods-10-03160-f002:**
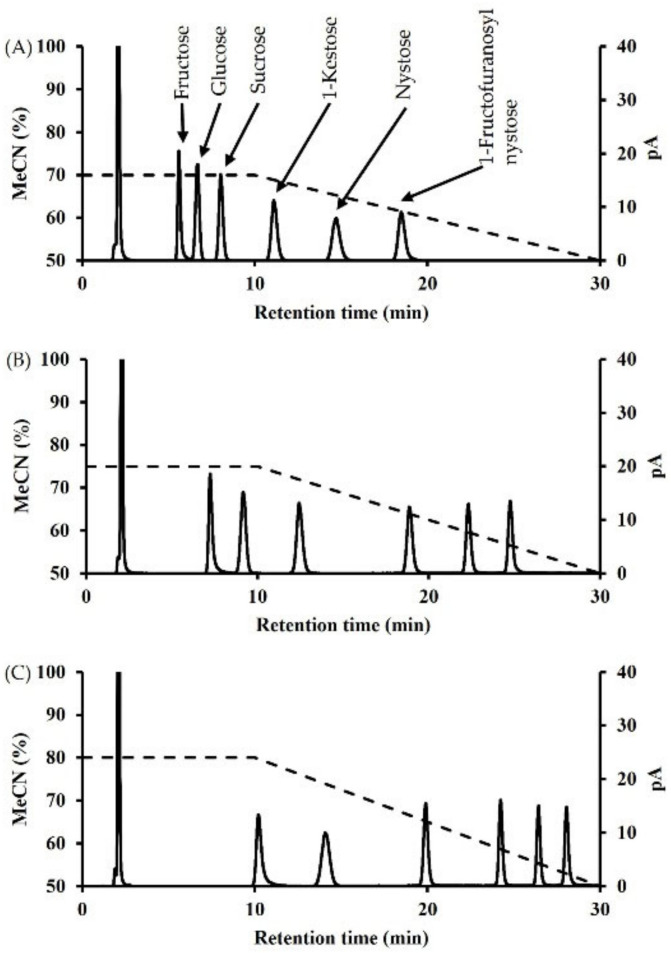
Chromatographic differences using saccharide standard by multi-gradient elution with (**A**) 70% acetonitrile, (**B**) 75% acetonitrile, and (**C**) 80% acetonitrile isocratic elution for 0 to 10 min, followed by reduction to 50% acetonitrile over 10 to 30 min.

**Figure 3 foods-10-03160-f003:**
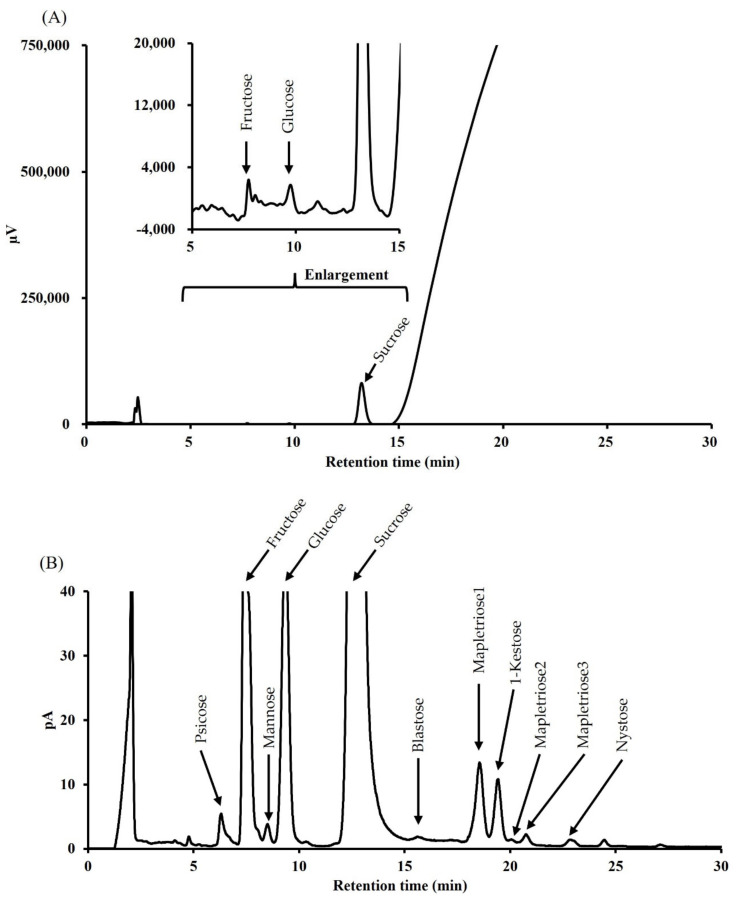
Comparison of the chromatograms of maple syrup at a common concentration of 10 mg/mL detected by RID (**A**) and CAD (**B**).

**Figure 4 foods-10-03160-f004:**
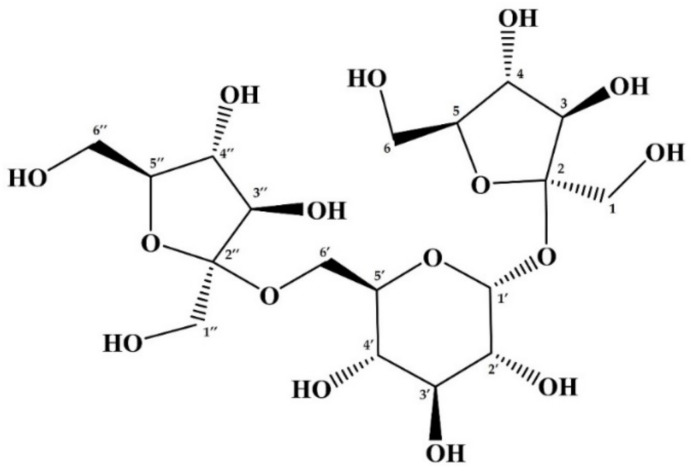
The structure of neokestose, purified and isolated from maple sap. The NMR chemical shift values for each number are shown in Table 3.

**Table 1 foods-10-03160-t001:** Validation data for the optimized analytical method using HILIC-CAD, including concentration range, linear equation, correlation coefficients, LOD, LOQ, intra- and inter-day precision, and accuracy.

Saccharides		Fructose	Glucose	Sucrose	1-Kestose	Nystose	Fructofuranosyl Nystose
Concentration range (µg/mL)		0.5–100	0.5−100	0.5−100	0.5−100	0.5−100	0.5−100
Linear equation		y = −0.0003x^2^ + 0.1193x + 0.0279	y = −0.0003x^2^ + 0.1268x + 0.0577	y = −0.0003x^2^ + 0.1237x + 0.0781	y = −0.0003x^2^ + 0.1149x + 0.0539	y = −0.0002x^2^ + 0.0980x + 0.0329	y = −0.0002x^2^ + 0.0995x + 0.0241
R^2^		0.9998	0.9998	0.9996	0.9998	0.9997	0.9999
LOD/LOQ (µg/mL)		0.25/0.5	0.25/0.5	0.25/0.5	0.25/0.5	0.25/0.5	0.25/0.5
Precision (%)	Intra-day (*n* = 9)	0.733	0.768	0.639	0.668	0.848	0.840
	Inter-day (*n* = 3)	0.630	1.46	1.02	1.67	1.46	1.71
Accuracy (%)		96.5	91.7	94.3	92.3	98.2	97.9

**Table 2 foods-10-03160-t002:** The concentration of saccharides in all grades of maple syrup under investigation.

Variables	Golden	Amber	Dark	Very Dark
Fructose (µg/10 mg)	21.3 ± 0.146	41.7 ± 0.295	65.6 ± 0.442	163 ± 1.46
Glucose (µg/10 mg)	31.6 ± 0.152	55.1 ± 0.396	83.3 ± 0.464	198 ± 2.12
Sucrose (mg/10 mg)	5.98 ± 0.159	5.94 ± 0.138	6.04 ± 0.115	5.49 ± 0.0506
Mapletriose1 ^(^*^)^ (µg/10 mg)	20.6 ± 1.23	19.6 ± 0.474	31.2 ± 0.491	40.6 ± 0.573
1-Kestose (µg/10 mg)	5.05 ± 0.357	10.1 ± 0.383	13.6 ± 0.142	25.3 ± 0.510
Nystose (µg/10 mg)	0.244 ± 0.0155	0.593 ± 0.00395	1.45 ± 0.0191	3.08 ± 0.0483

Data are presented as mean ± S.D. (*) The concentration of mapletriose1 was predicted using the calibration curve of 1-kestose.

**Table 3 foods-10-03160-t003:** ^1^H (800 MHz) and ^13^C (200 MHz)-NMR spectral data of neokestose in D_2_O.

Chemical Shift					
Residue	Position	δ_C_	δ_H_	*J* _H,H_	Type
Fruf β	1	64.15	3.64	-	s
	2	106.42–106.43	-	-	-
	3	78.92	4.2	8.9	d
	4	76.64	4.05	8.7, 8.7	dd
	5	84.07	3.87–3.89	-	m
	6	65.07	3.83	12.1, 11.9	dd
Glcp α	1′	94.72	5.39	3.9	d
	2′	73.73	3.55	3.9, 9.9	dd
	3′	75.15	3.73	9.6, 9.6	dd
	4′	71.89	3.51	9.9, 9.4	dd
	5′	74.25	3.91–3.94	-	m
	6′	63.02	3.77–3.79	-	m
Fruf β	1″	62.90	3.65	-	s
	2″	106.42–106.43	-	-	-
	3″	79.48	4.18	8.7	d
	4″	77.04	4.13	8.1, 8.5	dd
	5″	83.86	3.85–3.87	-	m
	6″	65.15	3.82	11.9, 12.1	dd

Chemical shifts (δ_C_ and δ_H_) and coupling constants (*J*
_H,H_) are shown in ppm and Hz, respectively.

## Data Availability

The data presented in this study are available in the article and its Appendix A.

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
