# Peer review of "Separation of Fructosyl Oligosaccharides in Maple Syrup by Using Charged Aerosol Detection"

_foods, 2021, doi:10.3390/foods10123160_

Round 1
Reviewer 1 Report
I have carefully reviewed the manuscript entitled “Separation of fructosyl oligosaccharides in maple syrup by using charged aerosol detection”. The optimized analytical method based on charged aerosol detection (CAD) in this study may be useful to determine the content of saccharides in maple syrup. However, this study is simple and some problems in this study should be resolved. The main comments are listed as following.
- Authors claimed that CAD has a wider detection range and higher resolution than the refractive index detection (RID). However, as stated in the Introduction, the RID detector is routinely used for saccharides analysis. The standards and samples should be analyzed by the RID and compared with the results obtained by CAD to confirm their statement.
- Generally, kestoses are categorized as 1-kestose, 6-kestose and neokestose. Can the analytical method based on the CAD analyze the 6-kestose?
- The author used 2D NMR to identify and analyze the structure of neokestose. The spectra of these 2D NMR should be given in the text or supplementary materials. ESI-MS can easily confirm the structure of neokestose, and the data should be provided.
- In the Abstract section, author stated that the neokestose has only been found in a few natural products. However, the neokestose can be produced by Xanthophyllomyces dendrorhous and Penicillium citrinum as shown in some studies.
Reviewer 2 Report
1) Although CAD seemed to be an efficient analytical method to measure FOS, it was not clear whether signals from CAD can be influenced by the presence of other phytochemicals in maple syrup. 2) Authors used a 30-min elution time for the CAD detection- is 30 min sufficient? Would other (longer) FOSs be detected at extended retention time? Minor questions: 1) Maple syrup used in this study was obtained from a local farm- what maple species are these maple syrup produced from? What grade is the maple syrup? Some definiton and standardization is needed. 2) English has to be polished.
Round 2
Reviewer 1 Report
Most questions have been addressed by the authors. However, I do not agree with the explanation about the comment 1. The author said that " While the RID method could only detect sucrose, glucose, and
fructose" and "Since the RID analytical method using could not performed gradient elution, the peaks of standard including relativity large saccharides could not be analyzed in a short time with sufficient separation." However, as shown in Figure 2 and 3, the saccharide standards and saccharide components of maple syrup were analyzed by a isocratic-gradient elution with acetonitrile. We do not think so. The chromatograms of the saccharide standards and saccharide components of maple syrup determined by the RID should be provided and compared with that in the Figure 3 under the same conditions. Thus, this paper may be more convincing. The legend of Figure 2 is error, for it is not isocratic-gradient elution.
